# Developing a localized resilience assessment framework for historical districts: A case study of Yazd, Iran

**Saeedeh Moayedfar**[1]*, **Hossein Mohebbi**[2], **Najmeh Mozaffaree Pour**[3], **Ayyoob Sharifi**[4]

**1** Department of Geography, Faculty of Human Sciences, Meybod University, Meybod, Iran, **2** Department of Management, Faculty of Human Sciences, Meybod University, Meybod, Iran, **3** Tallinn University of Technology, Academy of Architecture and Urban Studies, Tallinn, Estonia, **4** Hiroshima University, The IDEC Institute & Network for Education and Research on Peace and Sustainability (NERPS), Higashi Hiroshima City, Hiroshima, Japan

* moayedfar@meybod.ac.ir

## Abstract

Growing natural and man-made disasters necessitate enhanced resilience in urban historical districts, vital for cultural heritage and tourism. This study aims to develop a localized assessment framework tailored to the unique characteristics of Yazd, Iran, a UNESCO World Heritage site known for its ancient architecture and cultural significance. By adapting and downscaling indicators from established DROP and BRIC models, we evaluated resilience across seven key dimensions and 17 criteria. Using advanced multi-criteria decision-making methods, including Delphi, Fuzzy DEMATEL, Fuzzy ANP, and VIKOR, we prioritized and ranked the historical districts based on their resilience scores. The results revealed that the social dimension and housing infrastructure are the most crucial factors for resilience. Environmental and institutional dimensions, while important, were found to be less critical in comparison. The VIKOR analysis identified specific districts with lower adaptability, requiring targeted interventions. These findings provide valuable information for policymakers and urban planners, offering a robust framework for enhancing urban historical district resilience. This study provides a context-specific approach to resilience assessment, emphasizing the need for tailored strategies to preserve and strengthen the resilience of culturally significant urban areas.

## 1. Introduction

The rising frequency and intensity of adverse events in cities have propelled "urban resilience" to the forefront of scientific and policy discourse. This concept serves as an organizing principle, guiding actions and informing decision-making to create cities adaptable to multiple threats. However, a universally accepted definition of urban resilience remains elusive [1]. It is used in various disciplines, including crisis management, climate change, urban planning, building engineering [2–4]. Generally, it encompasses the capacity of cities to anticipate, withstand, recover from, and adjust to adverse events [5]. Cities are increasingly aware of their vulnerability and the need to build resilience. This requires a unified approach that addresses social, economic, and environmental aspects to create robust urban ecosystems [6], and it is

**Data availability statement:** All relevant data are within the paper.

**Funding:** The author(s) received no specific funding for this work.

**Competing interests:** The authors have declared that no competing interests exist.

emphasized that improving various planning, absorption, recovery, and adaptation capacities of cities is critical for their survival [7].

Historical districts embody a city's memory, showcasing its past through architecture, streetscapes, and unique heritage [8,9]. They significantly contribute to urban life by preserving cultural identity, attracting tourism and investment [10], promoting sustainability [10], fostering social bonds [11], and enhancing the quality of life. However, these districts face challenges. Balancing modern needs with historical preservation is crucial [12]. Understanding incremental urban changes and their impact on identity requires ongoing analysis [13]. Reconciling city competitiveness with cultural-religious identity necessitates careful consideration of regeneration projects [14]. Finally, managing evolving meanings and conflicting values within historic landscapes demands specific management policies [15].

Yazd, a historic city in Iran, is an exemplary case study for urban resilience regarding its status as a UNESCO World Heritage site [16] and its unique cultural and architectural heritage. Known for its ancient windcatchers, narrow alleys, and mud-brick buildings, Yazd embodies a rich historical tapestry that spans centuries. Its cultural significance and architectural uniqueness make it a vital subject for studying resilience in historical urban areas. However, recent urbanization poses several threats to Yazd's historical fabric: (i) management and economic challenges including lack of a comprehensive development plan, fragmented governance, limited private investment, and residents' financial constraints hinder progress, (ii) social and security issues like migration, social disharmony, displacement, and security concerns due to narrow streets and deteriorated buildings plague these areas [11], and (iii) physical and environmental problems such as mobility issues [17], building deterioration, inadequate services, earthquake vulnerability, climate change impacts, inappropriate interventions, and waste accumulation create challenges.

Evaluating resilience is crucial as historical districts like Yazd must preserve their heritage while facing these threats. This assessment should identify potential hazards such as earthquakes, pollution, and climate change [18] and gauge the capacity of these districts to withstand and adapt to such challenges. Additionally, it should also provide strategies to enhance their resilience.

This study aims to develop and pilot-test a framework for assessing urban resilience in Yazd, a historic Iranian city. We adapt and downscale the established DBRIC (District-based Baseline Resilience Indicators for Communities) and DROP (Disaster Resilience of Place-based) models [18] using a mixed-methods approach (Delphi, Fuzzy DEMATEL, Fuzzy ANP, VIKOR). The DROP model [19], with its 29 dimensions, assesses flexibility across environmental, social, financial, organizational, foundational, and community capacities. The BRIC model [20] incorporates diverse subsystems (environmental, economic, infrastructural, social, institutional, and community) with 49 indicators for a comprehensive resilience index. Social resilience [21,22] emphasizes a community's ability to navigate crises. Economic resilience assesses resourcefulness. Institutional resilience [23] focuses on adapting to changes. Infrastructural resilience examines the capacity of physical assets to withstand shocks. Community capital [19] explores neighborhood social cohesion.

This study deviates from past practices by employing a combination of these models to identify locally relevant resilience indicators for historic districts. Besides, decision-making models offer a structured and comprehensive framework for resilience assessment, facilitating scenario evaluation, stakeholder engagement, and iterative decision-making [24]. These valuable tools (despite limitations) enable informed decisions to enhance resilience.

The assessment results will assist municipalities, authorities, researchers, and the local community in preserving and managing urban fabric, making informed decisions on development, historic preservation, and risk management, and serving as a model for similar historical areas. Therefore, this study addresses urban resilience by providing a localized, context-specific

framework that addresses the unique challenges faced by historic districts. By integrating advanced multi-criteria decision-making techniques, the study offers novel insights and practical strategies for enhancing the resilience of culturally significant urban areas.

## 2. Literature review

### 2.1. Urban resilience

Holling (1973) introduced "resilience" in ecology, which later expanded to address long-term issues such as climate change [25] and short-term disasters [26]. Applications emerged in various fields, including natural hazards [27], social systems [28], human-environment systems [29], and social-ecological systems [30]. The 2005 Gothenburg Summit marked a shift in disaster planning from vulnerability reduction to building community resilience [6,31], reflecting a global prioritization of resilience over vulnerability reduction [32].

Urban resilience involves the capacity of an area to adapt and withstand shocks and disturbances while preserving essential functions and supporting the well-being of its residents [33]. It encompasses anticipating, absorbing, and recovering from disruptions and transforming in the face of long-term challenges like climate change [34]. Urban resilience integrates physical, social, economic, and environmental aspects to build sustainable and inclusive cities [35]. This dynamic, multidimensional process [35] strengthens communities by leveraging their capacities, leading to various definitions, approaches, and assessment models [36].

Urban resilience assessment typically employs two main approaches: indicator-based and place-based. Indicator-based methods measure specific metrics across infrastructure, social systems, economics, and the environment to assess a city's ability to withstand stressors [37,38]. These frameworks highlight areas for improvement and guide interventions. Place-based studies utilize models like BRIC, DBRIC, and TBRIC to assess resilience at the community level [19,39,40], considering both internal and external stressors and highlighting the importance of physical and socio-cultural aspects [41]. In their study, Liu et al. [42] develop a multidimensional urban resilience assessment system, examining social, economic, infrastructure, and ecological aspects, as well as spatiotemporal trends and spatial correlations.

Although numerous models of resilience have been proposed [28,43,44], the majority focus on the conceptual aspect of resilience rather than its quantitative measurement [31]. In contrast, models like BRIC (Baseline Resilience Indicators for Communities) and TBRIC (Tract Baseline Resilience Indicators for Communities) provide practical tools for measuring resilience in specific geographic contexts, offering a structured approach to assess resilience quantitatively [40].

### 2.2. Historical districts and resilience

Historical districts face unique challenges in maintaining resilience due to their cultural significance, aging infrastructure, and susceptibility to both natural and man-made hazards. The preservation of cultural heritage adds an additional layer of complexity to resilience planning and disaster management.

Research in Iran focuses on cities' ability to withstand and recover from environmental hazards like floods and earthquakes, extending beyond physical aspects to include social networks, economic stability, and governance. Studies employ diverse methods such as multi-criteria models for prioritizing resilience indicators [23], district-level resilience tracking [39], and context-specific analysis using methods like Smart PLS, ELECTRE, and F'ANP [45–47]. Social and physical vulnerability to earthquakes, especially in historic neighborhoods, is also a focus [48–50].

Beyond Iran, research on resilience in traditional commercial centers includes a framework for historical bazaars, considering tangible and intangible aspects (e.g., cultural, economic,

spatial) [51], and index-based models for performance assessment and resilience improvement [52]. Other studies explore resilience in various contexts like urban planning, energy, reconstruction, and capacity prioritization [53–55].

## 2.3. Resilience framework

Established frameworks like DROP and BRIC provide structured approaches to assessing resilience. These models offer valuable insights but have limitations, particularly in their applicability to the specific context of historic districts. These frameworks have been widely applied in urban resilience studies. However, their generalized nature often overlooks historical districts' unique characteristics and needs. This study aims to address these limitations by extracting, downscaling, and localizing indicators from these models to fit the urban district level of Yazd.

Multi-Criteria Decision Making (MCDM) Methods play a crucial role in resilience assessment and urban planning by allowing the evaluation and prioritization of various resilience factors. These methods help make informed decisions by considering multiple criteria and stakeholders' perspectives. Specifically, we employed Fuzzy DEMATEL (Decision Making Trial and Evaluation Laboratory) to identify and analyze the causal relationships among resilience factors, helping to understand their interdependencies, and Fuzzy ANP (Analytic Network Process) to prioritize resilience dimensions and criteria through pairwise comparisons and network modeling. Also, we utilized VIKOR to rank the historical districts based on their resilience scores and to identify areas needing targeted interventions.

Despite significant progress in urban resilience research in understanding its complexities, gaps remain in creating a unified theoretical framework, obtaining sufficient data, and generalizing results. This study acknowledges the inherent complexity of measuring urban resilience due to its multifaceted nature, spatial variations, and temporal dynamics [56,57]. While various urban resilience assessment methods exist, applying and identifying resilience criteria at the micro level, particularly for historical fabrics, remains challenging [39].

Accurate resilience assessment for historical areas is crucial. However, some aspects remain more challenging, including (i) global indicators in existing models often miss the unique characteristics of specific historical fabrics like those in Yazd, Iran, (ii) assessing the multifaceted nature of resilience, encompassing social, economic, environmental, and institutional aspects, is complex, and (iii) information on historical fabrics is often incomplete, scattered, and unreliable, making data collection difficult.

Therefore, there is a particular need for localized and context-specific resilience assessment frameworks for historical districts. This study addresses these gaps by combining indicator-based and place-based methods to assess resilience in Yazd's historical fabric. It provides a more nuanced understanding of local resilience indicators and employs novel qualitative (snowballing, Delphi) and quantitative (fuzzy DEMATEL and ANP) methods for a more precise assessment. By prioritizing historic districts with the VIKOR method, this research enables better planning to strengthen vulnerable areas, contributing significantly to the field of urban resilience.

## 3. Materials and methods

### 3.1. Study area

Yazd is one of Iran's ancient and historical cities, along with the Spice Road and Silk Road. The climate-friendly Iranian-Islamic architecture of the city is unique and is considered an important World Heritage Site. The historical structures in this city are diverse, and because of its 2500-year history, it is included in the UNESCO World Heritage [16].

Yazd's historical fabric consists of 13 districts with numerous historical and cultural monuments, making it a major tourist destination (Fig 1). The city's hot and dry climate is reflected

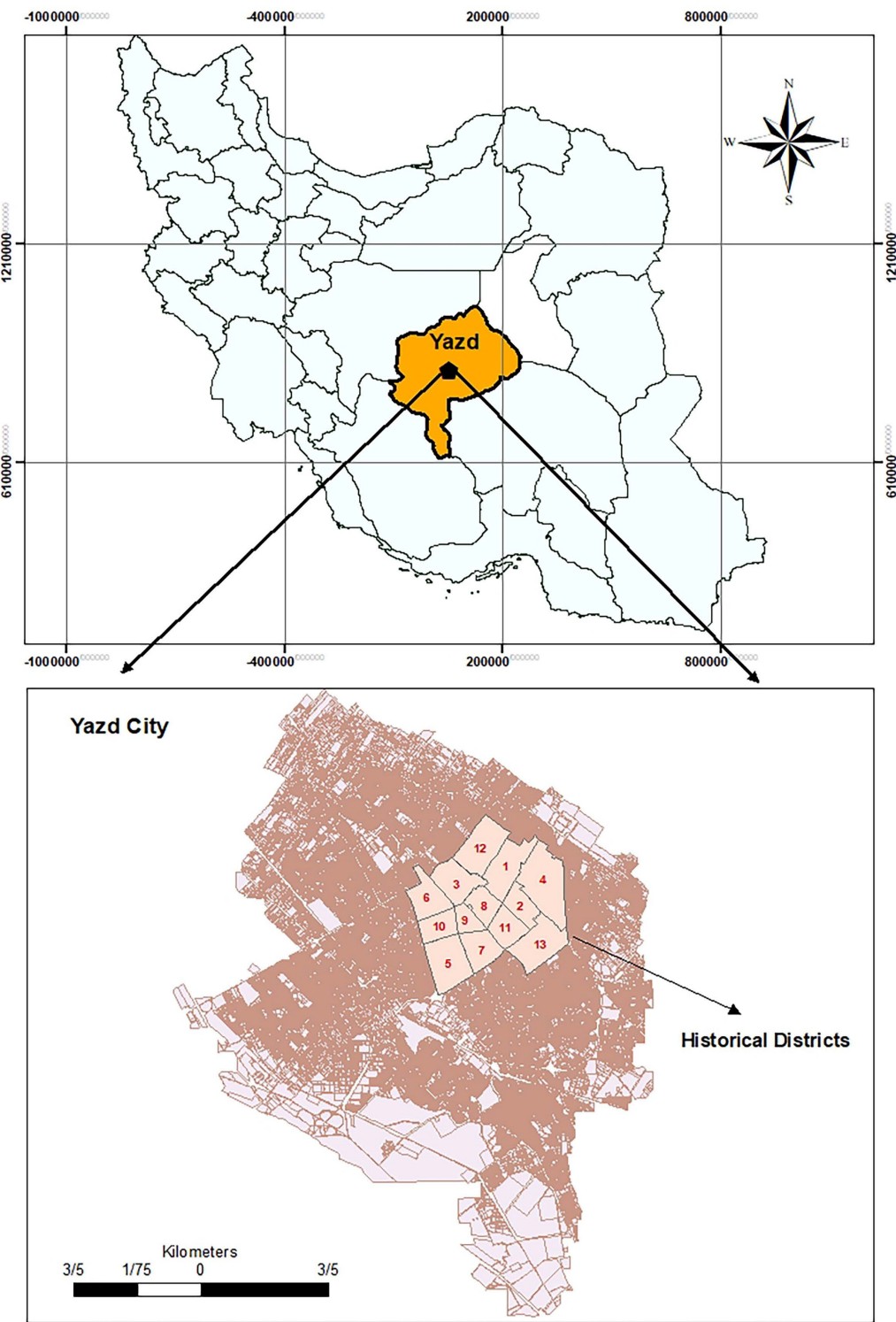

**Fig 1. Study area.** Map created by the authors using geospatial data provided by the Yazd Municipality Information and Communication Technology Organization (FAVA). The figure shows the location of Yazd within Iran and the historical districts of Yazd City. Published under the CC BY 4.0 license.

in its organically grown historical texture, well-adapted to the desert environment. This compact and integrated structure has contributed to the area's resilience. However, evaluating the resilience of its various components is crucial for effective planning and preparedness for future challenges.

### 3.2. Framework development

**3.2.1. Determination of a conceptual framework.** Cities, with their intricate systems, demand a focus on resilience. This dynamic process considers a community's inherent robustness and ability to adapt [58]. Consequently, a resilient approach is crucial for cities to effectively manage unforeseen shocks and stresses. Furthermore, historical neighborhoods within cities face additional challenges. This study adopts a conceptual framework to extract indicators from the DBRIC and DROP models (Fig 2).

Selecting relevant indicators is crucial when dealing with a wide range specific to the study area. Engaging subject-matter experts is a common approach for this purpose. In this study, a snowball sampling method identified 16 experts to form a panel. To align with research objectives, a set of criteria and indicators were established through a comprehensive literature review.

**3.2.2. Extracting dimensions of resilience indicators.** In this study, we utilized a resilience assessment framework with an indicator-based approach, focusing on the specificities of the 13 historical districts of Yazd. We applied consistent data obtained from national and local sources (Statistical Center of Iran and Yazd City ICT Organization) between 2021-2023 for a quantitative assessment of the 13 districts' resilience.

To achieve robust data collection, this study employed a two-step approach. First, a comprehensive literature review established the theoretical foundation. Second, snowball sampling, a non-probability technique, was utilized to recruit participants [59,60]. This method is

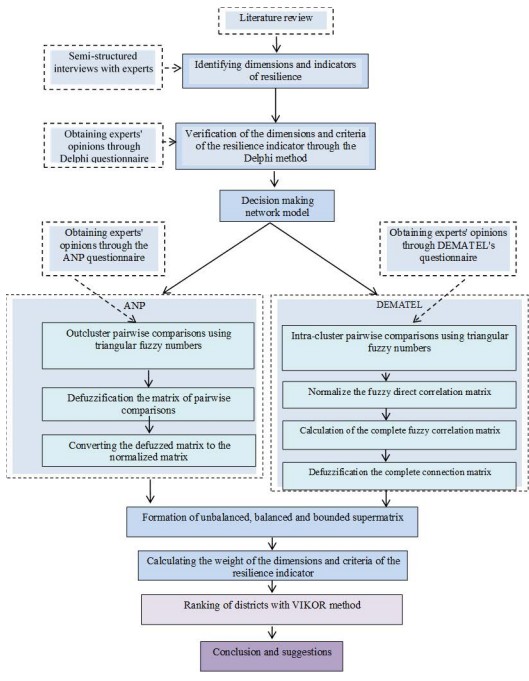

**Fig 2. Stepwise methodological approach in this study.**

particularly effective when dealing with hard-to-reach populations or sensitive topics. Following snowball sampling principles, 16 individuals were selected for semi-structured interviews. Selection criteria prioritized expertise in urban planning, specifically crisis and resilience in urban historical areas. The participants comprised a diverse group, including 4 managers, 9 municipal experts from the historical districts, and 3 university professors. Notably, over half possessed over 10 years of experience, and their academic backgrounds ranged from bachelor's degrees (31%) to master's (50%) and Ph.D.s (19%) (Fig 3).

The interview process employed a combination of pre-determined and open-ended questions. Standardized questions focused on the DROP and BRIC models' specific dimensions and resilience criteria (Table 1) to assess their applicability in the studied localities. Additionally, open-ended questions explored new resilience measures relevant to the historical fabric of Yazd. This two-pronged approach aimed to gather comprehensive data for a nuanced understanding of resilience in the local context.

**3.2.3. Verification of resilience indicators.** This study employed the Delphi technique, a multi-round expert consultation process, to refine the initial list of 34 resilience criteria derived from the DROP and BRIC models [75]. The Delphi method leverages expert knowledge and fosters collective intelligence through iterative rounds of questionnaires and feedback.

Round 1 The initial round involved soliciting expert input on the criteria. This resulted in the modification of the list through the addition of dimensions, elimination of redundant criteria, and integration of related ones.

Round 2 Experts assessed the revised list of seven dimensions and 23 criteria using a 5-point Likert scale (strongly agree to disagree strongly). Criteria with an average score below 3 (indicating low agreement) were removed, resulting in a final list of 17 criteria. Internal consistency of the ratings was validated using Kendall's reliability and correlation coefficient (0.774 and 0.704, respectively).

Round 3 In the final round, the 16 experts ranked the remaining 17 variables based on their impact on Yazd's historical districts' resilience (1 = least impact, 5 = most impact). Experts were again provided with average scores from the previous round. The findings, presented in Table 5, confirmed all sub-criteria. Kendall's reliability and correlation coefficient for this round were 0.814 and 0.762, respectively, demonstrating strong agreement among experts.

This iterative Delphi process facilitated the refinement of the resilience criteria, ensuring their relevance and applicability to the specific context of Yazd's historical fabric.

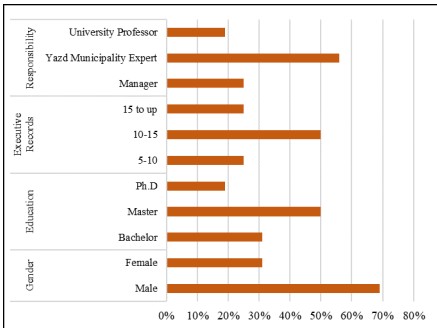

**Fig 3. The descriptive information in the study sample (N = 16).**

**Table 1. The selected set of indicators for six dimensions of disaster resilience (Compiled from various sources).**

| Aspects | Factors | Description | Efficacy | Justification/Inspiration |
|---|---|---|---|---|
| **Social Resilience** | Transportation access | % car ownership | Positive | [23] |
| | people < 65 years old | Percent non-elderly population (percentage of the population < 65 years old) | Positive | [40,61,62] |
| | Level of literacy | % literate people | Positive | [63,64] |
| | sex rate | Number of men to women | Positive | [23,39] |
| | health insurance | % people with health insurance | Positive | [47] |
| | Population density | Number of people per unit area | Negative | [65,66] |
| | Public recreational facilities | % sport hall, cinemas in each district | Positive | [47] |
| **Economic Resilience** | Total economic participation | Percentage of active persons to the total population | Positive | [64] |
| | Employment rate | Percent employed | Positive | [19] |
| | dependency burden | The ratio of the inactive population to the working population | Negative | [67] |
| | Housing capital | Average land value | Positive | [19] |
| | Women Employment | Percentage of working women | Positive | [45] |
| | Commercial infrastructure | Ratio Commercial infrastructure in each district | Positive | [47] |
| | Dependence on tourism sector | %The tourism sector in each district | Positive | [20,23] |
| **Institutional Resilience** | Municipal services | Percent municipal services in the district | Positive | [20] |
| | Population changes | Population changes over time | Negative | [68,69] |
| **Housing/ Infrastructure** | Temporary shelter availability | The number of accommodation centers per square meter | Positive | [23,70] |
| | Durable houses | Percentage of sustainable houses | Positive | [63] |
| | School restoration potential | Number of schools per unit area | Positive | [71] |
| | Fire stations | The number of routes to the fire station | Positive | [63,72] |
| | Police stations | The number of routes to the Police stations | Positive | [47] |
| | Healthcare houses | The number of routes to the health centers | Positive | [55] |
| | Medical capacity | The number of routes to the Hospital | Positive | [73] |
| | Occupancy level | Built up area of land | Negative | [61] |
| | Houses upper 100 square meters | % residential units more than 100 square meters | Positive | [47] |
| | Public cultural centers | Percentage of Public cultural land use | Positive | [23] |
| | Housing lifetime | Percent housing units built under 30 years old | Positive | [20,27] |
| | Access routes | Access to the ways | Positive | [66] |
| **Environmental** | City parks and urban Green spaces | Percentage of Park and Green spaces | Positive | [23,74] |
| | Urban Facilities | Percentage of Facilities | Positive | [66] |
| | People in the housing | Number of people per household | Negative | [65,66] |
| **Community Capital** | Migrants to districts | % The population that is a national or international migrant | Negative | [20,61] |
| | Cultural and heritage services | % Active tourism centers in the district | Positive | [47] |
| | Religious centers | % Religious centers in the district | Positive | [47,73] |

### 3.3. Multi-criteria decision making (MCDM) methods

In order to obtain the relationships caused by the dependence between the dimensions and criteria and their weight (importance), we used the hybrid method of DEMATEL and ANP in fuzzy mode. Since the factors in the research have internal dependence and mutual influence on each other, the best method to measure the relationships between them is DEMATEL, and on the other hand, to decide the ultimate weight of each dimentions and criteria we utilized ANP. The combination of these two methods is as follows:

First, by using the DEMATEL method, we determined the internal relationships between dimensions and criteria and the process of defuzzification and normalization. This matrix forms a part of the unbalanced supermatrix of the ANP. Therefore, according to Table 2, W22

and W33 matrixes are calculated using the DEMATEL method, and W21 and W32 matrixes are calculated using the ANP method.

**(a) Fuzzy-DEMATEL.** Introduced in 1973, the DEMATEL technique [76] offers a structured approach to analyzing complex decision-making scenarios. By creating a model that evaluates the influence relationships between various criteria [77], DEMATEL helps visualize these interdependencies. This is particularly beneficial in "fuzzy environments" with subjective judgments [66] where fuzzy logic, introduced by Zadeh [78], can be integrated. Fuzzy logic utilizes membership values between 0 and 1 [79] to represent the complexities of expert opinions, potentially leading to a more nuanced understanding of cause-and-effect relationships within the decision-making framework. Additionally, DEMATEL may simplify the process by reducing the number of criteria needed for effective agent evaluation.

To implement the Fuzzy- DEMATEL technique in this research, we performed the following steps:

1. Creating a group of experts in order to collect their group knowledge to solve the problem

2. Determining the evaluation criteria as well as designing language scales

3. Creating the initial direct correlation fuzzy matrix by gathering expert opinions (to measure the relationship between the criteria, we should put them in a square matrix and ask the experts to compare them in pairs based on their impact on each other. In this survey, experts will express their opinions based on Table 3).

4. Direct association fuzzy matrix normalization (A linear scaling transformation is used as normalization).

$$\tilde{a}_{ij} = \sum_{j=1}^{n} \tilde{Z}_{ij} = \left( \sum_{j=1}^{n} l_{ij}, \sum_{j=1}^{n} m_{ij} \sum_{j=1}^{n} r_{ij} \right) \text{and } r = \max_{1 \leq i \leq n} \left( \sum_{j=1}^{n} r_{ij} \right) \tag{1}$$

$$\tilde{X} = \begin{bmatrix} \tilde{X}_{11} & \tilde{X}_{12} & \cdots & \tilde{X}_{1n} \\ \tilde{X}_{21} & \tilde{X}_{22} & \cdots & \tilde{X}_{2n} \\ \vdots & \vdots & \ddots & \vdots \\ \tilde{X}_{m1} & \tilde{X}_{m2} & \cdots & \tilde{X}_{mn} \end{bmatrix} \text{and } \tilde{X}_{ij} = \frac{\tilde{Z}_{ij}}{r} = \left( \frac{l_{ij}}{r}, \frac{m_{ij}}{r}, \frac{r_{ij}}{r} \right) \tag{2}$$

**Table 2. The common frame of the supermatrix.**

|  | Goal | Dimensions | Criteria |
|---|---|---|---|
| **Goal** | I |  |  |
| **Dimensions** | W$_{21}$ | W$_{22}$ |  |
| **Criteria** |  | W$_{32}$ | W$_{33}$ |

**Table 3. The scale of Fuzzy linguistic: [80].**

| Linguistic terms | Influence score | Triangular fuzzy number |
|---|---|---|
| **No influence** | 0 | (0, 0, 0.25) |
| **Very low influence** | 1 | (0, 0.25, 0.50) |
| **Low influence** | 2 | (0.25, 0.50, 0.75) |
| **High influence** | 3 | (0.50, 0.75, 1) |
| **Very high influence** | 4 | (0.75, 1, 1) |

5. Calculation of the total correlation fuzzy matrix

$$\left[l_{ij}^{''}\right] = X_l \times \left(I - X_l\right)^{-1} \qquad \left[m_{ij}^{''}\right] = X_m \times \left(I - X_m\right)^{-1} \qquad \left[r_{ij}^{''}\right] = X_r \times \left(I - X_r\right)^{-1} \qquad (3)$$

Overall, to enhance the analysis's reliability, Fuzzy DEMATEL, identifies the most critical criteria for resilience assessment. The methodology employs a two-step procedure. First, the overall objective and a list of criteria, comprising seven dimensions and 17 specific criteria, are established. Subsequently, the direct influence matrix is constructed. Experts compare each pair of criteria using a five-point fuzzy scale (0 = no influence, 1 = low influence, 2 = normal influence, 3 = high influence, 4 = very high influence) (details in Table 3). Equations 4 and 5 are utilized to normalize and fuzzify the matrix based on the CFCS strategy. This procedure is replicated to determine the internal relationships among the 17 criteria.

$$K = \frac{1}{\max\limits_{1 \le i \le n} \sum_{j=1}^{n} a_{ij}}, i, j = 1, 2 \dots n \qquad\qquad N = K \text{x} A \qquad (4)$$

$$x_{ij}^{crisp} = \frac{l_{ij}^{''} - l_{ij}^{''} \times l_{ij}^{''} + u_{ij}^{''} \times u_{ij}^{''}}{1 - l_{ij}^{''} + u_{ij}^{''}} \qquad x_{ij} = x_{ij}^{crisp} = L + x_{ij}^{crisp} \times \Delta. \qquad (5)$$

**(b) Fuzzy-ANP.** The Analytical Network Process (ANP) offers a robust method for evaluating complex decision-making scenarios with multiple variables [81]. Unlike traditional hierarchical approaches, ANP allows for interdependent relationships between variables within a network structure [82,83]. This makes it particularly suitable for addressing the intricate connections often found in fields like community resilience [84]. The ANP model typically involves a hierarchy with a central goal, multiple criteria, and various alternatives [85].

This study employed the ANP to analyze 17 variables identified through the Delphi method, forming a network with one goal, seven criteria, and full connectivity between all elements. The methodology, as outlined by Eskandari et al. [86], involves constructing the ANP network, pairwise comparisons to weight criteria, defining internal dependence within criteria using a fuzzy scale (Table 4), and finally calculating overall weights for sub-criteria based on linguistic variables [87].

Following the within-cluster analysis, Fuzzy DEMATEL is employed to compare criteria across different clusters (dimensions). This step identifies the external effects between dimensions and criteria using expert opinions. Similar to the previous stage, verbal expressions and triangular fuzzy numbers (refer to Table 4) are used for pairwise comparisons in fuzzy ANP.

**Table 4. Language scales for trouble and significance: [86].**

| Linguistic scale for difficulty | Linguistic scale for importance | Triangular fuzzy scale | Triangular fuzzy reciprocal scale |
|---|---|---|---|
| **Just equal** | Just equal | (1, 1, 1) | (1, 1, 1) |
| **Equally difficult (ED)** | Equally importance (EI) | (1/2, 1, 1/2) | (2/3, 1, 1) |
| **Weakly more difficult (WMD)** | Weakly more importance (WMI) | (1, 3/2, 2) | (1/2, 2/3, 1) |
| **Strongly more difficult (SMD)** | Strongly more importance (SMI) | (3/2, 2, 5/2) | (2/5, 1/2, 2/3) |
| **Very strongly more difficult (VSMD)** | Very strongly more importance(VSMI) | (2, 5/2, 3) | (1/3, 2/5, 1/2) |
| **Absolutely more difficult (AMD)** | Absolutely more importance (AMI) | (5/2, 3, 7/2) | (2/7, 1/3, 2/5) |

Overall, the process involves several steps:

1. Expert Assessment: 16 experts evaluate the seven dimensions' objectivity using a fuzzy scale. An acceptable consistency ratio (CR) below 0.1 confirms the results' validity [88] using the Equation 6:

$$CI(A) = \frac{1}{n(n-1)} \sum_{i=1}^{n} \sum_{j=1}^{n} \left| a_{ij} \frac{w_j}{w_i} - 1 \right|$$ (6)

2. Defuzzification: The fuzzy matrix is converted into a crisp matrix using Equation 7:

$$X = \frac{L + (4M) + U}{6}$$ (7)

3. Weighting Dimensions: The weight of each dimension is calculated from the transformed matrix.

4. Comparing Criteria and Supermatrix Construction: Experts perform pairwise comparisons of the 17 criteria across all seven dimensions and Sub-matrices representing relationships within and between levels are combined to create a supermatrix.

**(c) VIKOR.** VIKOR, developed by Opricovic (1998), is a prominent Multi-Criteria Decision Making (MCDM) method based on a cumulative metric LP function [89]. Its primary focus lies in ranking and selecting the most suitable alternative from a set of options. The VIKOR method follows a structured approach:

1. Determine the best (fi*) and worst (fi^) values for all criterion functions, where i = 1, 2,..., n. If the $i^{th}$ function is a benefit, fi* is calculated as the maximum value among fij (j = 1,..., J), and fi^ is the minimum value. If the i-th function is a cost, fi* is the minimum value, and fi^ is the maximum value.

2. Compute the values of Sj (which represents the weighted and normalized Manhattan distance) and Rj (j = 1, 2,..., J) (which represents the weighted and normalized Chebyshev distance) using the following formulas (Equation 8):

$$Sj = sum\left[ \frac{wi(fi * - fij)}{fi * - fi^{\wedge}}, i = 1, \ldots, n \right]$$

$$Rj = max\left[ \frac{wi(fi * - fij)}{fi * - fi^{\wedge}}, i = 1, \ldots, n \right]$$ (8)

Here, wi represents the weights of the criteria, expressing the relative significant of the criteria according to the decision maker's preferences.

3. Compute the values of Qj (j = 1, 2,..., J) using the formula (Equation 9), where S* = min(Sj, j = 1,..., J), S^= max(Sj, j = 1,..., J), R* = min(Rj, j = 1,..., J), R^ = max(Rj, j = 1,..., J).

$$Q_j = v\left[ \frac{S_j - S^*}{S^{\wedge} - S^*} \right] + (1-v)\left[ \frac{R_j - R^*}{R^{\wedge} - R^*} \right]$$ (9)

The weight v represents the strategy for maximizing group utility (typically set at 0.5), whereas 1-v represents the weight assigned to individual regret. To incorporate the criterion related to R in S, the value of v can be adjusted to (n + 1)/(2n), where n is the number of criteria.

4. The alternatives will be ranked in ascending order based on the quantities of S, R, and Q. This process will produce three separate ranking lists (Equation 10):

$$S_j = \sum_{i=1}^{n} w_i \cdot \frac{f_i^* - f_{ij}}{f_i^* - f_i^-}; \qquad R_j = \max_i \left[ w_i \cdot \frac{f_i^* - f_{ij}}{f_i^* - f_i^-} \right] \qquad (10)$$

5. The compromise solution achieved using VIKOR is deemed acceptable by decision makers as it offers maximum utility for the majority (indicated by the minimum value of S) and minimum individual regret for the opponent (indicated by the minimum value of R). The measures S and R are combined with Q to determine a compromise solution, which serves as the foundation for reaching an agreement through mutual concessions [90].

## 4. Results

### 4.1. Resilience criteria results

Employing the Delphi technique, we evaluated relevant indicatorsthrough a series of three rounds with a panel of experts. This iterative process resulted in the identification seven dimensions and 17 criteria essential for assessing the resilience of historical districts in Yazd.

Considering the local focus of this research, a selection process refined the initial criteria, resulting in a final model with seven key dimensions for assessing historical district resilience in Yazd (Table 5). These dimensions encompass environmental variables,

**Table 5. The initial list reduction by the Delphi method.**

| Dimension | Symbol | Criteria | Symbol | Descriptive Statistics | | | |
|---|---|---|---|---|---|---|---|
| | | | | Mean | Standard Deviation | Maximum | Minimum |
| Social | $C_1$ | Migrants to districts | S1 | 3.5 | 1.0041 | 5 | 1 |
| | | literacy rate | S2 | 3.7 | 1.0212 | 5 | 2 |
| | | sex ratio | S3 | 3.1 | 1.1593 | 4 | 1 |
| Environmental | $C_2$ | City parks and urban Green spaces | S4 | 3.3 | 1.4194 | 4 | 1 |
| | | Urban Facilities | S5 | 4 | 0.9428 | 5 | 2 |
| Institutional | $C_3$ | Population stability | S6 | 3.8 | 1.2292 | 5 | 2 |
| | | Cultural and heritage services | S7 | 4.2 | 0.7032 | 5 | 3 |
| Housing and Infrastructure | $C_4$ | Aid stations | S8 | 3.6 | 0.9826 | 5 | 1 |
| | | Durable houses | S9 | 3.9 | 1.1635 | 5 | 2 |
| | | Houses upper 100 square meters | S10 | 3.7 | 0.8432 | 5 | 2 |
| Community Capital | $C_5$ | Religious centers | S11 | 3.1 | 1.3142 | 4 | 1 |
| | | School restoration potential | S12 | 3.4 | 1.2425 | 5 | 1 |
| Social and Infrastructure | $C_6$ | Population density | S13 | 3.7 | 0.8164 | 5 | 2 |
| | | Healthcare centers | S14 | 3.6 | 1.7131 | 5 | 1 |
| Economic | $C_7$ | Subsistence load | S15 | 3.8 | 0.8755 | 5 | 2 |
| | | Total economic participation | S16 | 3.4 | 1.1352 | 5 | 1 |
| | | Employment rate | S17 | 3.8 | 1.1005 | 5 | 1 |

housing and infrastructure, economic aspects, social dynamics, and institutional frame-works. Two environmental variables reflect the urban environment's ability to absorb shocks. Three housing and infrastructure variables assess the physical assets' response and recovery capabilities. Three economic variables evaluate the community's resourceful-ness and strength. Three social variables examine the inherent social capacity within and between districts. Two institutional variables focus on planning and adapting to environ-mental changes.

Then, after evaluation of seven dimentions by experts, the initial matrix (calculated using DEMATEL) is incorporated (Table 5).

Since the dimensions and criteria in the research model exhibit internal dependencies and mutual influences, the fuzzy DEMATEL method was applied to evaluate the internal rela-tionships among them. Additionally, to determine the final weights of each dimension and criterion in light of their interconnections, the fuzzy ANP method was utilized. The integra-tion of these two methods, as shown in Table 2, was conducted as follows: initially, using the fuzzy DEMATEL method, the internal relationships among dimensions ($W_{22}$) and criteria ($W_{33}$) were established based on expert opinions. Next, the resulting matrix was defuzzified and normalized, forming a part of the unweighted supermatrix in the fuzzy ANP framework. Another segment of the unweighted supermatrix was derived using the fuzzy ANP method, in which pairwise comparisons between dimensions and criteria ($W_{21}$ and $W_{32}$) were conducted according to expert evaluations.

Finally, following the stages of the fuzzy ANP method, the unweighted supermatrix was converted into a weighted supermatrix, and the final weights of the resilience dimensions and criteria were determined, as presented in (Table 6).

Table 6 and Fig 4 reveal that the Housing and Infrastructure dimension is the most crucial factor for resilience in Yazd's historical districts. This aligns with the potential deterioration issues and outdated infrastructure in these neighborhoods. The Community Capital dimension also holds significant weight, likely due to residents' active social partic-ipation. The top three criteria are Durable houses, Aid stations, and Urban Facilities. This

**Table 6. Final weight and ranking of resilience dimensions and criteria in Yazd historical districts.**

| Dimension | Symbol | Weight | Ranking | Criteria | Symbol | Weight | Ranking |
|---|---|---|---|---|---|---|---|
| **Social** | $C_1$ | 0.131 | 5 | Migrants to districts | S1 | 0.077 | 5 |
| | | | | literacy rate | S2 | 0.046 | 12 |
| | | | | sex ratio | S3 | 0.044 | 13 |
| **Environmental** | $C_2$ | 0.136 | 4 | City parks and urban Green spaces | S4 | 0.032 | 17 |
| | | | | Urban Facilities | S5 | 0.081 | 3 |
| **Institutional- community** | $C_3$ | 0.102 | 7 | Population stability | S6 | 0.072 | 6 |
| | | | | Cultural and heritage services | S7 | 0.057 | 8 |
| **Housing and Infrastructure** | $C_4$ | 0.189 | 1 | Aid stations | S8 | 0.087 | 2 |
| | | | | Durable houses | S9 | 0.088 | 1 |
| | | | | Houses upper 100 square meters | S10 | 0.080 | 4 |
| **Community Capital** | $C_5$ | 0.174 | 2 | Religious centers | S11 | 0.054 | 9 |
| | | | | School restoration potential | S12 | 0.041 | 15 |
| **Social and Infrastructure** | $C_6$ | 0.110 | 6 | Population density | S13 | 0.036 | 16 |
| | | | | Healthcare centers | S14 | 0.052 | 10 |
| **Economic** | $C_7$ | 0.158 | 3 | Subsistence load | S15 | 0.043 | 14 |
| | | | | Total economic participation | S16 | 0.047 | 11 |
| | | | | Employment rate | S17 | 0.063 | 7 |

emphasizes the importance of prioritizing improvements in housing quality, establishing more aid centers, and expanding urban facilities to strengthen resilience and sustainability in these historical areas.

## 4.2. Ranking of historical districts in Yazd based on resilience indicators with VIKOR

Following the weight determination for dimensions and criteria, a decision matrix is constructed (Table 7). This matrix includes 13 historical districts evaluated across 17 resilience criteria.

Since the criteria have different units, the matrix is normalized. This involves calculating the maximum (fj*) and minimum (fj-) values for each criterion. Next, utility values (S) and regret values (R) are computed for each district based on the normalized data (Table 7).

These values are then used to calculate the VIKOR index (Q) and the parameter V reflects the decision-makers' preferences regarding the balance between utility and regret values. A lower Q value indicates better overall performance for a district based on the combined criteria. Table 8, represents the calculation of Q for each district, ultimately ranking them based on their resilience. This process is repeated for each dimension. Finally, all districts are comprehensively ranked by combining the rankings across all dimensions (Table 9). This final ranking provides a holistic assessment of each district's resilience, considering all relevant factors.

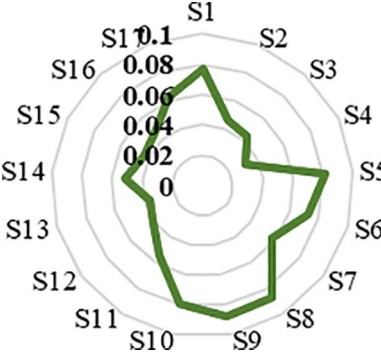

**Fig 4.  Final individual indicators' weight obtained from the Fuzzy DEMATEL and ANP.**

**Table 7.  Values of Rj and Sj in districts (All criteria).**

| No. District | Districts | $S_j$ | $R_j$ | No. District | Districts | $S_j$ | $R_j$ |
|---|---|---|---|---|---|---|---|
| 1 | Fahadan | 0.552 | 0.079 | 8 | Mosala | 0.554 | 0.079 |
| 2 | Gazorgah | 0.552 | 0.082 | 9 | Poshtebagh | 0.660 | 0.080 |
| 3 | Ghaharmonar | 0.700 | 0.078 | 10 | Saredorah | 0.614 | 0.087 |
| 4 | Gonbadesabz | 0.669 | 0.079 | 11 | Sheshbadgiri | 0.703 | 0.075 |
| 5 | Khoramshad | 0.538 | 0.080 | 12 | Shykhdad | 0.618 | 0.080 |
| 6 | Koochebeyook | 0.599 | 0.081 | 13 | Zartoshtiha | 0.701 | 0.088 |
| 7 | Mollafarajolla | 0.459 | 0.082 | | | | |

**Table 8. Q scores of areas within the seven resilience aspects.**

| No. District | Urban District | Social Resielance | Environmental Resielance | Institutional- com- munity Resielance | Housing & Infra- structure Resielance | Community Capi- tal Resielance | Social & Infrastruc- ture Resielance | Economic Resielance |
|---|---|---|---|---|---|---|---|---|
| 1 | Fahadan | 0.597 | 0 | 0.7189 | 0.560 | 0.664 | 0.514 | 0.712 |
| 2 | Gazorgah | 0.437 | 0.853 | 0.0742 | 0.676 | 0.606 | 0.705 | 0.184 |
| 3 | Gha- harmonar | 0.681 | 0.924 | 0.761 | 0.498 | 0.745 | 0.441 | 0.735 |
| 4 | Gonbades- abz | 0.842 | 0.884 | 0.954 | 0.409 | 0.712 | 0.474 | 1 |
| 5 | Khoram- shad | 0.523 | 0.954 | 0.774 | 0.314 | 0.911 | 0.271 | 0.119 |
| 6 | Koochebe- yook | 0.453 | 0.973 | 0.363 | 0.660 | 0.912 | 0.381 | 0.150 |
| 7 | Mollafara- jolla | 0 | 0.865 | 0.615 | 0.698 | 0.742 | 0 | 0.507 |
| 8 | Mosala | 0.912 | 0.913 | 0.839 | 0.268 | 0 | 0.527 | 0.529 |
| 9 | Poshtebagh | 0.706 | 0.949 | 0.833 | 0.346 | 0.645 | 0.628 | 0.378 |
| 10 | Saredorah | 0.699 | 0.853 | 0.869 | 0.886 | 0.789 | 0.438 | 0.211 |
| 11 | Sheshbadgiri | 0.713 | 0.909 | 0.994 | 0.460 | 0.631 | 0.499 | 0.653 |
| 12 | Shykhdad | 0.897 | 0.669 | 0 | 0.431 | 0.715 | 1 | 0.981 |
| 13 | Zartoshtiha | 0.861 | 0.978 | 0.856 | 0.831 | 0.976 | 0.454 | 0.671 |

**Table 9. Combination resilience value and the districts ranking.**

| No. District | Districts | Q | Ranking | No. District | Districts | Q | Ranking |
|---|---|---|---|---|---|---|---|
| 1 | Fahadan | 0.076 | 3 | 8 | Mosala | 0.078 | 4 |
| 2 | Gazorgah | 0.079 | 5 | 9 | Poshtebagh | 0.159 | 9 |
| 3 | Ghaharmonar | 0.187 | 12 | 10 | Saredorah | 0.133 | 8 |
| 4 | Gonbadesabz | 0.165 | 10 | 11 | Sheshbadgiri | 0.186 | 11 |
| 5 | Khoramshad | 0.067 | 2 | 12 | Shykhdad | 0.127 | 7 |
| 6 | Koochebeyook | 0.114 | 6 | 13 | Zartoshtiha | 1 | 13 |
| 7 | Mollafarajolla | 0.009 | 1 | | | | |

Fig 5 shows that roughly 25% of the districts exhibit stronger social capital resilience compared to others. This can be linked to factors like higher literacy rates, public awareness, a balanced sex ratio, and population density. Conversely, the southern districts, particularly Zartoshtiha, Koochebeyook, and Mollafarajolla, appear most vulnerable regarding social capital variables. While social capital disparities are evident, some districts also show vulner-abilities in Environmental and Institutional-Community dimensions. These aspects warrant targeted interventions to improve resilience.

Approximately 40% of the districts demonstrate high to moderate levels of economic capital resilience. Regarding infrastructure, 54% of the districts show moderate resilience, with a stronger performance observed in the social-infrastructural integration dimension. Overall, 62% of the districts exhibit some level of resilience across all dimensions. The resilience status of the districts is presented in detail and in general based on the Q values, categorized into three levels: high resilience (Q < 0.1), medium resilience (0.1 < Q < 0.15), and low resilience (Q > 0.15), in Figs 5–7.

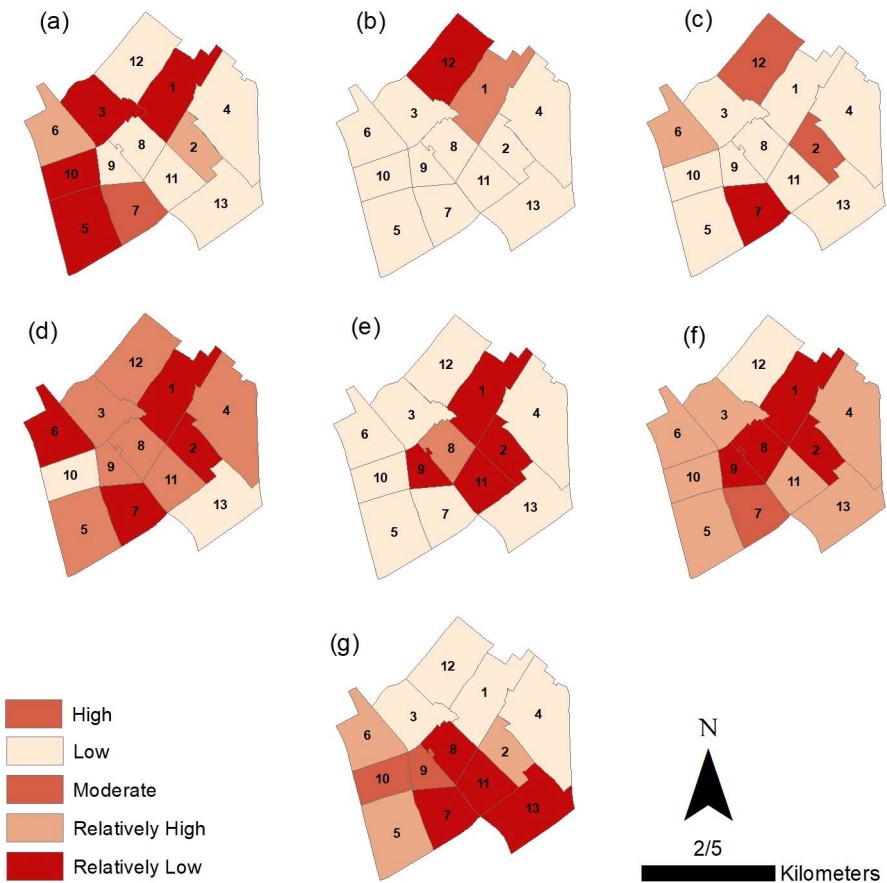

**Fig 5. The resilience value of Yazd historical areas (a) Social, (b) Environmental, (c) Institutional- community, (d) Housing and infrastructural, (e) Community Capita, (f) Social & Infrastructure, (g) Economic.** Map created by the authors using geospatial data provided by the Yazd Municipality Information and Communication Technology Organization (FAVA) and processed information from the manuscript. Published under the CC BY 4.0 license.

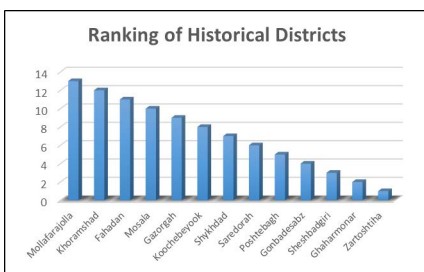

**Fig 6. VIKOR values for the 13 historical areas and their ranking.**

## 5. Discussion

This study aimed to develop a comprehensive framework for assessing urban resilience specifically in historical districts, using Yazd, Iran, as a case study. By implementing a comprehensive, multi-criteria decision-making methods, this study assesses resilience through seven key dimensions covering structural, social, economic, and environmental factors that

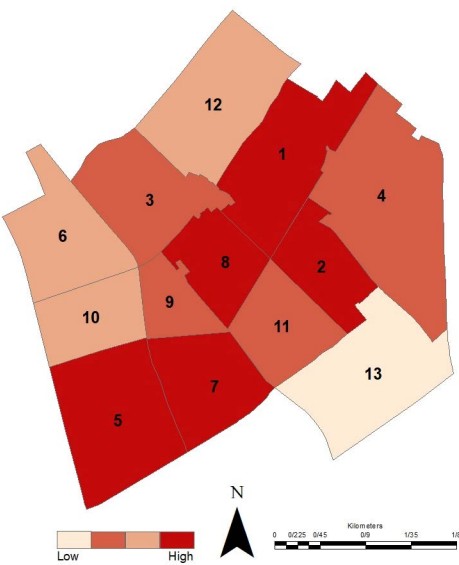

**Fig 7. The spatial results of the resilience of Yazd historical districts in integrated dimensions scores.** Map created by the authors using geospatial data provided by the Yazd Municipality Information and Communication Technology Organization (FAVA) and processed information from the manuscript. Published under the CC BY 4.0 license.

collectively shape resilience in such unique urban settings. The findings of this study align with the broader literature on urban resilience, particularly in historical districts. For example, cities in Italy, France, and Greece with dense historical architecture, such as Turin, Strasbourg, and Athens, like Yazd, possess a very ancient history and consequently showcase magnificent features in the architecture and urban planning of their historical urban centers. At the same time, they face similar challenges regarding resilience, where implementing policies that prioritize the renewal of infrastructure in historical districts can ensure safety and functionality while preserving cultural heritage [91]. Besides, in their studies, Cutter et al. [5] and Shi et al. [7] emphasize the importance of integrating social, economic, and physical dimensions to build resilient urban ecosystems. Here, the results offer several key insights and practical implications for enhancing the resilience of historical districts:

## 5.1. Significance of housing and infrastructure

The findings underscore the critical importance of the Housing and Infrastructure dimension, which emerged as the most important factor for resilience in Yazd's historical districts. This aligns with the challenges of deterioration and outdated infrastructure commonly faced by historical areas. Durable houses, aid stations, and urban facilities were identified as the top three criteria, emphasizing the need for targeted improvements in these areas. Enhancing housing quality and expanding urban facilities are essential strategies for strengthening resilience and ensuring the sustainability of historical districts. This study's emphasis on housing and infrastructure corroborates findings from Khalil [12], who noted the critical role of maintaining historical buildings to preserve urban heritage.

## 5.2. Role of community capital

Community Capital also held significant weight in the resilience assessment, reflecting the active social participation of residents in Yazd. Social cohesion, cultural and heritage services,

and population stability were among the key criteria within this dimension. The high ranking of these factors highlights the vital role that social networks and community engagement play in resilience. Policymakers and urban planners should focus on fostering social bonds and community-driven initiatives to enhance resilience. The role of community capital and social infrastructure identified in this study is consistent with the literature highlighting social cohesion and community engagement as pivotal for resilience [6,11]. This study extends these insights by providing a localized assessment in the context of Yazd's unique cultural heritage.

### 5.3. Environmental and institutional dimensions

The Environmental dimension, including urban green spaces and environmental health, was another critical area. Although it ranked lower than Housing and Infrastructure they remain critical, particularly given the increasing impacts of climate change. The relatively lower emphasis on environmental resilience contrasts with some studies prioritizing environmental factors [34]. However, this divergence can be attributed to the specific challenges faced by historical districts, where the immediate concern often revolves around preserving the built environment and cultural heritage [8].

Similarly, the Institutional-Community dimension, which encompasses governance, planning, and adaptation strategies, is essential for coordinated and effective resilience efforts. Strengthening these areas through comprehensive policies and adaptive management practices is crucial for long-term resilience.

### 5.4. Economic resilience

Economic aspects, such as employment rate and economic participation, also play a pivotal role in resilience. The study found that approximately 40% of the districts demonstrate high to moderate levels of economic resilience. Economic stability and resourcefulness are foundational for recovery and adaptation in the face of adverse events. Strategies to boost local economies, support small businesses, and create job opportunities are vital for enhancing economic resilience.

### 5.5. Spatial variations in resilience

Using the VIKOR analysis, the study reveals significant spatial variations in resilience across historical districts, showing that districts like Fahadan and Khoramshad exhibited higher resilience, whereas others like Zartoshtiha and Mollafarajolla were more vulnerable. These spatial disparities highlight the need for targeted interventions tailored to each district's specific needs and vulnerabilities. By addressing these unique challenges, planners can ensure more equitable and effective resilience strategies across the entire city.

The framework developed in this study offers a globally adaptable tool for assessing resilience in historical urban districts. By integrating environmental, economic, social, and institutional resilience dimensions, it provides a structured model that can benefit urban planners and policymakers worldwide: (i) **Application in infrastructure improvement:** Cities with historical districts that suffer from aging or inadequate infrastructure can benefit from prioritizing investments in durable housing and urban facilities to enhance structural resilience. Implementing policies that prioritize infrastructure renewal in historical districts can ensure safety and functionality while preserving cultural heritage. (ii) **Enhancing community engagement:** Community engagement as a significant resilience factor, underscors the need for strengthening social infrastructure by fostering community engagement and supporting cultural heritage services. In regions where cultural ties and community identity play a critical role in collective resilience, engaging local communities in resilience planning

can substantially improve adaptability to both environmental and economic shocks. (iii) **Environmental resilience management:** The framework suggests implementing targeted environmental management strategies to mitigate climate-related risks and improve adaptive capacities. Resilience frameworks could integrate green spaces and environmental health measures to mitigate risks related to urban heat and flooding. By doing so, historical districts can enhance adaptive capacities to climate change, preserving both heritage and livability. (iv) **Institutional coordination:** This study underscores the need for institutional collaboration framework to improve planning, governance, and disaster preparedness. In cities like Yazd, where resilience efforts often overlap due to historical and modern administrative complexities, adopting an integrated framework with coordinated institutional efforts can enhance resilience, and (v) **Economic resilience:** In districts where economic stability directly influences resilience, fostering local businesses, and creating job opportunities can improve adaptive capacity. Economic initiatives that support small enterprises and provide employment in tourism or heritage management contribute to resilience, allowing these communities to recover quickly from adverse events.

While this study provides a robust framework for resilience assessment, it has limitations. The reliance on expert judgment in the Delphi and DEMATEL methods may introduce bias, and the complexity of the decision-making models may pose challenges for practical implementation. However, due to the lack of existing models, this methodology provides valuable insights because it systematically examines a wide range of indicators encompassing various aspects of resilience. Furthermore, it utilizes various decision-making techniques to prioritize dimensions and analyze the spatial distribution of resilience changes at the neighborhood level. Future research should aim to refine these methods, explore the framework's applicability in different cultural and geographical contexts, and develop user-friendly tools to facilitate broader adoption. Longitudinal studies are also recommended to monitor resilience changes over time and assess the long-term impact of implemented strategies.

## 6. Conclusion

Resilience assessment is crucial for effectively managing historical districts, particularly balancing tourism and preservation. This study on Yazd, a UNESCO World Heritage site, identifies and prioritizes key resilience dimensions through expert input and advanced multi-criteria decision-making techniques. Despite some data limitations, the study provides valuable insights for further research and policy development.

The study reveals that social and economic dimensions are the most significant factors influencing resilience of historical districts. The analysis shows an uneven distribution of resilience across different neighborhoods, with some areas exhibiting higher vulnerability. The most critical resilience factors identified include durable housing, aid stations, and urban facilities, highlighting the importance of infrastructure improvements and social capital enhancement.

Based on the findings, the study recommends several policy actions to enhance the resilience of historical districts:

- **Targeted strategies:** Develop specific strategies tailored to the unique needs of historical areas to bolster their resilience.

- **Educational programs:** Implement educational programs to raise awareness about resilience and preservation among residents and stakeholders.

- **Stakeholder collaboration:** To ensure cohesive and effective resilience-building efforts, Foster collaboration among local authorities, residents, and other stakeholders.

- **Technological integration:** Utilize technology for monitoring interventions and managing resilience improvements.

- **Balanced approach:** Seek to balance preservation efforts with resilience improvements to maintain the historical and cultural integrity of the districts.

Future research should focus on expanding this framework to other historical districts and cultural contexts to validate its applicability and generalize its findings. Longitudinal studies are needed to monitor resilience changes and assess the long-term impact of resilience-building interventions. Further refinement of indicators and methods, incorporating more diverse data sources, will enhance the robustness and accuracy of resilience assessments. Integrating resilience research with other disciplines, such as urban planning, environmental science, and social sciences, can foster more holistic and effective approaches to urban resilience.

## Author contributions

**Conceptualization:** Saeedeh Moayedfar, Ayyoob Sharifi.

**Formal analysis:** Saeedeh Moayedfar, Hossein Mohebbi.

**Methodology:** Saeedeh Moayedfar, Hossein Mohebbi.

**Visualization:** Saeedeh Moayedfar.

**Writing – original draft:** Saeedeh Moayedfar.

**Writing – review & editing:** Najmeh Mozaffaree Pour, Ayyoob Sharifi.

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
