## [Decision Letter · Decision Letter 0]

17 Oct 2024

PONE-D-24-27356Developing a Localized Resilience Assessment Framework for Historical Districts: A Case Study of Yazd, IranPLOS ONE

Dear Dr. Moayedfar,

Thank you for submitting your manuscript to PLOS ONE. After careful consideration, we feel that it has merit but does not fully meet PLOS ONE’s publication criteria as it currently stands. Therefore, we invite you to submit a revised version of the manuscript that addresses the points raised during the review process.

We look forward to receiving your revised manuscript.

Kind regards,

Ziqiang Zeng, Ph.D.

Academic Editor

PLOS ONE

Journal requirements: When submitting your revision, we need you to address these additional requirements. 1. Please ensure that your manuscript meets PLOS ONE's style requirements, including those for file naming. The PLOS ONE style templates can be found at https://journals.plos.org/plosone/s/file?id=wjVg/PLOSOne_formatting_sample_main_body.pdf and https://journals.plos.org/plosone/s/file?id=ba62/PLOSOne_formatting_sample_title_authors_affiliations.pdf 2. Please note that PLOS ONE has specific guidelines on code sharing for submissions in which author-generated code underpins the findings in the manuscript. In these cases, all author-generated code must be made available without restrictions upon publication of the work. Please review our guidelines at https://journals.plos.org/plosone/s/materials-and-software-sharing#loc-sharing-code and ensure that your code is shared in a way that follows best practice and facilitates reproducibility and reuse. 3. We note that you have indicated that there are restrictions to data sharing for this study. PLOS only allows data to be available upon request if there are legal or ethical restrictions on sharing data publicly. For more information on unacceptable data access restrictions, please see http://journals.plos.org/plosone/s/data-availability#loc-unacceptable-data-access-restrictions.  Before we proceed with your manuscript, please address the following prompts: a) If there are ethical or legal restrictions on sharing a de-identified data set, please explain them in detail (e.g., data contain potentially identifying or sensitive patient information, data are owned by a third-party organization, etc.) and who has imposed them (e.g., a Research Ethics Committee or Institutional Review Board, etc.). Please also provide contact information for a data access committee, ethics committee, or other institutional body to which data requests may be sent. b) If there are no restrictions, please upload the minimal anonymized data set necessary to replicate your study findings to a stable, public repository and provide us with the relevant URLs, DOIs, or accession numbers. For a list of recommended repositories, please seehttps://journals.plos.org/plosone/s/recommended-repositories. You also have the option of uploading the data as Supporting Information files, but we would recommend depositing data directly to a data repository if possible. We will update your Data Availability statement on your behalf to reflect the information you provide. 4. We note that your Data Availability Statement is currently as follows: [All relevant data are within the manuscript and its Supporting Information files.] Please confirm at this time whether or not your submission contains all raw data required to replicate the results of your study. Authors must share the “minimal data set” for their submission. PLOS defines the minimal data set to consist of the data required to replicate all study findings reported in the article, as well as related metadata and methods (https://journals.plos.org/plosone/s/data-availability#loc-minimal-data-set-definition). For example, authors should submit the following data: - The values behind the means, standard deviations and other measures reported;- The values used to build graphs;- The points extracted from images for analysis. Authors do not need to submit their entire data set if only a portion of the data was used in the reported study. If your submission does not contain these data, please either upload them as Supporting Information files or deposit them to a stable, public repository and provide us with the relevant URLs, DOIs, or accession numbers. For a list of recommended repositories, please see https://journals.plos.org/plosone/s/recommended-repositories. If there are ethical or legal restrictions on sharing a de-identified data set, please explain them in detail (e.g., data contain potentially sensitive information, data are owned by a third-party organization, etc.) and who has imposed them (e.g., an ethics committee). Please also provide contact information for a data access committee, ethics committee, or other institutional body to which data requests may be sent. If data are owned by a third party, please indicate how others may request data access.

Reviewers' comments:

Reviewer's Responses to Questions

**Comments to the Author**

1. Is the manuscript technically sound, and do the data support the conclusions?

Reviewer #1: Yes

Reviewer #2: Yes

2. Has the statistical analysis been performed appropriately and rigorously? 

Reviewer #1: Yes

Reviewer #2: Yes

3. Have the authors made all data underlying the findings in their manuscript fully available?

Reviewer #1: Yes

Reviewer #2: Yes

4. Is the manuscript presented in an intelligible fashion and written in standard English?

Reviewer #1: Yes

Reviewer #2: Yes

5. Review Comments to the Author

Reviewer #1: Thank you for this interesting research. This study aims to develop a localized assessment framework tailored to the unique characteristics of heritage site known for its ancient architecture and cultural significance. Hereby are some comments that may help you improve on it:

1. Please check if these data are correct. Some data in Table 8 are 0, and the Sj is 1.116 of Zartoshtiha in Table 8.

2. Please discuss more about the results and show more contributions of this research and explain how the research results can be used to give values to different area in the world.

I hope that these notes are helpful in reviewing your article.

Reviewer #2: This manuscript developed a comprehensive framework to assess resilience for Historical Districts. 7 key dimensions and 17 cirteria were integrated into the framework. Overall, this manuscript is well ogrinized and technical sound. Some concerns should be addressed before further production.

1.Since this manuscript mergerd many dimensions and criteria into the framework, please explain how the weights of different parameter determined.

2.The study area is quite small. Is this framework proposed by this manuscript suitable for maybe large area. Please explain in detail.

3.For resilience assessment, the following references are suggested to be included.

https://doi.org/10.1016/j.ijdrr.2023.103649

https://doi.org/10.1016/j.uclim.2023.101591

https://doi.org/10.1016/j.scs.2023.105109

4.English presentation should be strengthen.

6. PLOS authors have the option to publish the peer review history of their article (what does this mean? ). If published, this will include your full peer review and any attached files.

**Do you want your identity to be public for this peer review?** For information about this choice, including consent withdrawal, please see our Privacy Policy .

Reviewer #1: No

Reviewer #2: No

---

## [Author Response · Author response to Decision Letter 1]

21 Nov 2024

Dear Editor,

Thank you for giving us the opportunity to submit a revised draft of the manuscript Number: PONE-D-24-27356 entitled “Developing a Localized Resilience Assessment Framework for Historical Districts: A Case Study of Yazd, Iran” for consideration for publication in the Journal of PLOS ONE.

We sincerely appreciate the constructive feedback provided to our manuscript. Based on these valuable suggestions, we have made substantial revisions to the manuscript, addressing all of the major points raised by reviewers. Below, we briefly summarize the main revisions:

Response to Reviewer 1:

1. Data accuracy: The Sj value in Table 8 was corrected to address a typographical error. The Q scores in Table 8 have been clarified, indicating that a score of 0 reflects optimal performance for the respective neighborhood dimension (Page 15).

2. Expanded discussion: We enriched the discussion on the broader applicability of our findings, particularly regarding resilience strategies in historical cities with similar architecture and urban challenges, such as Turin, Strasbourg, and Athens (Pages 17-19).

Response to Reviewer 2:

1. Parameter weights: We added details about the methodology used to assign weights to various criteria, highlighting this in Tables 5 and 6 (Page 13).

2. Framework applicability: We clarified that our resilience assessment framework is adaptable to larger areas by detailing a phased approach for regional evaluations and outlining technological aids for data management at scale.

3. Additional references: Per the reviewer’s suggestion, we have incorporated additional references to strengthen the resilience assessment background (Page 3).

4. English clarity: The manuscript’s language has been revised for clarity and coherence throughout.

Besides, we changed the reference style to the Vancouver style.

The revised manuscript file with tracked changes and reviewers respond are attached. Thank you for your continued interest in our research.

Sincerely,

Prof. Saeedeh Moayedfar, Prof. Hossein Mohebbi, Dr. Najmeh Mozaffaree Pour, Prof. Ayyoob Sharifi

---

## [Decision Letter · Decision Letter 1]

22 Dec 2024

Developing a Localized Resilience Assessment Framework for Historical Districts: A Case Study of Yazd, Iran

PONE-D-24-27356R1

Dear Dr. Moayedfar,

We’re pleased to inform you that your manuscript has been judged scientifically suitable for publication and will be formally accepted for publication once it meets all outstanding technical requirements.

Kind regards,

Ziqiang Zeng, Ph.D.

Academic Editor

PLOS ONE

Additional Editor Comments (optional):

Reviewers' comments:

Reviewer's Responses to Questions

**Comments to the Author**

1. If the authors have adequately addressed your comments raised in a previous round of review and you feel that this manuscript is now acceptable for publication, you may indicate that here to bypass the “Comments to the Author” section, enter your conflict of interest statement in the “Confidential to Editor” section, and submit your "Accept" recommendation.

Reviewer #1: All comments have been addressed

Reviewer #2: All comments have been addressed

2. Is the manuscript technically sound, and do the data support the conclusions?

Reviewer #1: Yes

Reviewer #2: Yes

3. Has the statistical analysis been performed appropriately and rigorously? 

Reviewer #1: Yes

Reviewer #2: Yes

4. Have the authors made all data underlying the findings in their manuscript fully available?

Reviewer #1: Yes

Reviewer #2: No

5. Is the manuscript presented in an intelligible fashion and written in standard English?

Reviewer #1: Yes

Reviewer #2: Yes

6. Review Comments to the Author

Reviewer #1: First, the authors have incorporated additional references to strengthen the resilience assessment background. Second, the manuscript’s language has been revised for clarity and coherence throughout. Overall, the comments have been addressed. I have no further comment.

Reviewer #2: The manuscript is technically sound, and the data support the conclusions. All my concerns have been well addressed. Thank you.

7. PLOS authors have the option to publish the peer review history of their article (what does this mean? ). If published, this will include your full peer review and any attached files.

**Do you want your identity to be public for this peer review?** For information about this choice, including consent withdrawal, please see our Privacy Policy .

Reviewer #1: No

Reviewer #2: No

---

## [Editor Report · Acceptance letter]

PONE-D-24-27356R1

PLOS ONE

Dear Dr. Moayedfar,

I'm pleased to inform you that your manuscript has been deemed suitable for publication in PLOS ONE. Congratulations! Your manuscript is now being handed over to our production team.

Kind regards,

on behalf of

Dr. Ziqiang Zeng

Academic Editor

PLOS ONE